# Assessing Diagnosis of *Candida* Infections: A Study on Species Prevalence and Antifungal Resistance in Northern Morocco

**DOI:** 10.3390/jof10060373

**Published:** 2024-05-23

**Authors:** Islam Ahaik, Juan Carlos Nunez-Rodríguez, Jamal Abrini, Samira Bouhdid, Toni Gabaldón

**Affiliations:** 1Laboratoire de Chimie et Microbiologie Appliquées et Biotechnologies, Faculté des Sciences, Université Abdelmalek Essaâdi, Tétouan 93000, Morocco; islam.ahaik@irbbarcelona.org (I.A.); sbouhdid@uae.ac.ma (S.B.); 2Institute for Research in Biomedicine (IRB Barcelona), The Barcelona Institute of Science and Technology, Baldiri Reixac, 10, 08028 Barcelona, Spain; 3Barcelona Supercomputing Centre (BSC-CNS), Plaça Eusebi Güell, 1-3, 08034 Barcelona, Spain; 4Catalan Institution for Research and Advanced Studies (ICREA), 08010 Barcelona, Spain; 5CIBER de Enfermedades Infecciosas, Instituto de Salud Carlos III, 28029 Madrid, Spain

**Keywords:** *Candida* infections, diagnostics, antifungal resistance, *Candida metapsilosis*

## Abstract

The incidence of *Candida* infections has increased in the last decade, posing a serious threat to public health. Appropriately facing this challenge requires precise epidemiological data on species and antimicrobial resistance incidence, but many countries lack appropriate surveillance programs. This study aims to bridge this gap for Morocco by identifying and phenotyping a year-long collection of clinical isolates (n = 93) from four clinics in Tetouan. We compared the current standard in species identification with molecular methods and assessed susceptibility to fluconazole and anidulafungin. Our results identified limitations in currently used diagnostics approaches, and revealed that *C. albicans* ranks as the most prevalent species with 60 strains (64.52%), followed by *C. glabrata* with 14 (15.05%), *C. parapsilosis* with 6 (6.45%), and *C. tropicalis* with 4 (4.30%). In addition, we report the first identification of *C. metapsilosis* in Morocco. Susceptibility results for fluconazole revealed that some isolates were approaching MICs resistance breakpoints in *C. albicans* (2), and *C. glabrata* (1). Our study also identified anidulafungin resistant strains in *C. albicans* (1), *C. tropicalis* (1), and *C. krusei* (2), rendering the two strains from the latter species multidrug-resistant due to their innate resistance to fluconazole. These results raise concerns about species identification and antifungal resistance in Morocco and highlight the urgent need for more accurate methods and preventive strategies to combat fungal infections in the country.

## 1. Introduction

The growing incidence of infections caused by human fungal pathogens is a global health problem of major concern [1]. It is estimated that 6.5 million cases of invasive fungal infections occur each year in the world, resulting in 3.8 million deaths, of which 2.5 million were directly attributable to the infection [2]. The limited classes of antifungals available for clinical use, coupled with the growing emergence of drug and multi-drug resistant strains, is making the situation even more complicated, reducing treatment options and worsening patient outcomes.

Infections caused by species of the *Candida* genus (Candidiasis) rank among the most common fungal infections worldwide. Moreover, during the recent coronavirus pandemic, COVID-19-associated candidiasis (CAC) was noted as one of the serious complications of hospitalized patients [3]. *Candida* is a paraphyletic genus that includes over 200 species, of which some are considered opportunistic human pathogens [4]. Five *Candida* species rank as the top etiological agents of candidiasis, being collectively responsible for more than 90% of the global incidence: *Candida albicans*, *Candida glabrata*, *Candida parapsilosis*, *Candida tropicalis*, and *Candida krusei* [5], with the relative contribution of each species varying across geographical regions. Furthermore, the Centers for Diseases Control and Prevention (CDC) recently issued a warning about a newly emerged species, *Candida auris*, due to its rapid spread, long-term persistence, and multidrug resistance properties [6]. More recently, the World Health Organization (WHO) considers *C. albicans*, *C. glabrata*, *C. parapsilosis*, *C. tropicalis*, and *C. auris* to be major threats to public health and listed them within the critical and high priority groups, which emphasizes the need for prevention strategies such as continuous surveillance of their incidence and resistance [1].

Despite the relevance of *Candida* infections, we still have very little information about the relative incidence of the different pathogenic species or their resistance profiles in most countries, particularly those from less favored regions such as the African continent [2].

The Global Action For Fungal Infections (GAFFI) recently estimated that 3,300,000 persons in Morocco (9% of the population) suffer from a fungal infection each year [7]. Moreover, recurrent vulvovaginal candidiasis is considered as the second most frequent fungal infection in the country, with an incidence rate of 2794 per 100,000 females. The identification of *Candida* species in Morocco relies mostly on phenotypic methods rather than molecular ones. Although phenotypic approaches save time and costs, the accuracy of the results remains low when it comes to identifying a particular species [8]. As a result, Morocco often represents a gap in global fungal pathogen and antimicrobial resistance surveillance programs [9,10], which prevents the development of informed clinical policies.

The aim of this study is to gain a first insight on the prevalence of *Candida* species in Tetouan, a northern city of Morocco with an approximate population of 400,000. For this we compared routine phenotypic methods and molecular tools by applying them to a one-year-long strain collection (n = 93) from four clinical laboratories. Our results identified nine different *Candida* species, including the first reported case of *Candida metapsilosis* in Morocco, and revealed common limitations of phenotypic methods.

## 2. Materials and Methods

### 2.1. Clinical Samples

A one-year sample collection was performed starting from March 2021 to February 2022. Ninety-three isolates of *Candida* were obtained from different patients and several anatomical sites (Table 1). These were collected from four different clinical laboratories in Tetouan, Morocco. All the samples were delivered in Sabouraud Dextrose Agar (SDA) plates, with metadata indicating the clinical origin. One of the laboratories additionally provided some strains in both SDA and Brilliance^TM^
*Candida* (Thermo Scientific^TM^, Waltham, MA, USA).

### 2.2. Germ Tube Test

The samples were classified by the corresponding clinical laboratory staff based on the ability of the fungal cells to produce germ tube, a key pathogenicity factor in *C. albicans.* For this, after isolation from various clinical origins, the strains were grown on SDA plates. The next day, a small portion of a pure colony from each sample was inoculated in human serum and incubated at 37 °C for 3 h. The results were observed using light microscopy. Two groups were distinguished: germ tube-positive strains (named GTT+) were those capable of producing a short, non-septic filament, and germ tube-negative strains (named GTT−) were those that did not.

### 2.3. Culture on CHROMagar^TM^ Candida

In order to evaluate the reliability of the germ tube test, we identified the strains using the chromogenic medium CHROMagar^TM^ Candida (Paris, France). This medium gives a presumptive identification based on the colony’s appearance. CHROMagar contains per litter agar (15 g), peptone (10 g), Chromogenic mix (22 g), and chloramphenicol (0.5 g), with a pH of 6.1. The medium was prepared according to the manufacturer’s instructions, being heated to boiling until completely dissolved (autoclaving is not required) and then poured to sterile Petri plates. Subsequently, the samples were inoculated and incubated at 35 °C for 48 h. All isolates plated on the chromogenic media were judged by colony morphology and pigmentation in accordance with the manufacturer’s instructions.

### 2.4. Yeast Colony PCR

The molecular analysis was performed using a combination of two fungal-specific oligonucleotide primers: ITS1 (forward primer) [5′-TCC GTA GGT GAA CCT GCC-3′] and NL4 (reverse primer) [5′-GGTCCGTGT TTCAAGACGG-3′]. The total reaction volume was 40 µL in a 96 well microplate. Each well contained 20 µL of 2X Taq Master Mix (Dongsheng Biotech Co., Guangzhou, China), a concentration of 0.4 µM of both forward and reverse primers, and 16 µL of water. Using pipette tips, a unique fungal colony (mid-log phase) of each sample was put in the mixture, and *C. glabrata* CBS 138 was used as positive control. The process was conducted in a thermal cycler (SimpliAmp, Thermo Fisher, Waltham, MA, USA) for one initial step of 5 min at 94 °C, followed by 29 cycles of 30 s at 94 °C, 30 s at 55 °C, and 90 s at 72 °C. The final cycle was 5 min at 72 °C. PCR products were then analyzed by electrophoresis in a 1.5% agarose gel run at 90 V for 45 min.

### 2.5. Sanger Sequencing and BLAST Identification

PCR products were purified using QIAquick PCR purification kit (Qiagen, Hilden, Germany). Later, the purified DNA of all the strains were sequenced by Macrogen, and the results were compared to the reference sequences available in the Genbank database using the BLAST search tool (https://blast.ncbi.nlm.nih.gov/Blast.cgi, accessed on 1 December 2022).

### 2.6. Antifungal Susceptibility Testing

Antifungal susceptibility to fluconazole (FLC) (Sigma-Aldrich, St. Louis, MO, USA) and anidulafungin (ANI) (Pfizer Pharmaceutical Group, New York, NY, USA) representing, respectively, the azoles and echinocandins antifungal classes, was assessed according to European Committee of Antimicrobial Susceptibility Testing broth microdilution method (EUCAST E.Def 7.3.2). The final antifungal concentrations ranged from 0.25–16 µg/mL for FLC and 0.015–8 µg/mL for ANI. The microplates were then incubated for 24 h at 35 °C. Afterwards, the results were revealed using a microplate reader at a wavelength of 530 mn. The isolates were then categorized as susceptible, intermediate, or resistant based on their minimal inhibitory concentration (MIC_50_), which represents the concentration required to inhibit 50% of the growth of planktonic yeasts, following the clinical breakpoints established by EUCAST.

## 3. Results

### 3.1. Species Identification

The initial screening test performed by the clinical laboratories staff indicated that 54 out of 93 (58%) isolates were germ tube-positive *Candida* species, whereas germ tube-negative species comprised 39 isolates (42%). According to these results and to the general assumption that *C. albicans* is the major germ-tube-forming *Candida*, *C. albicans* was the most abundant species. However, germ tube tests cannot differentiate between *C. albicans* and *C. dubliniensis*, both germ-tube-forming, and nor can they distinguish among non-*albicans* species. Consequently, we utilized more specific methods to achieve a more precise identification of the collected strains.

We first plated all strains on CHROMagar Candida, which is a color-based identification method. The observed colony morphologies mostly corresponded to the ones described by the manufacturer (Figure 1); i.e., bluish to light green colonies for *C. albicans*, mauve colonies for *C. glabrata*, dark metallic blue colonies for *C. tropicalis,* and fuzzy pink colonies for *C. krusei.* Of note, the manufacturer instructions do not mention expected colors for other *Candida* species. Besides these ones, and as noted in earlier studies, we observed other colors on this medium (pale yellow, dark purple), and the colonies rendering those colors remained unidentified (Figure 1 right panel indicates observed colors).

We finally used a molecular identification approach based on ITS amplification by PCR coupled to Sanger sequencing and followed by comparison to sequence databases (see Section 2). This approach identified 11 different species among the 93 isolates, and revealed some incongruences with the above mentioned morphology-based identification approaches (Figure 1, Table 1). *C. albicans* was the most abundant species with 60 strains (64.52%) followed by *C. glabrata* (14 strains, 15.05%), *C. parapsilosis* (6, 6.45%), and *C. tropicalis* (4, 4.30%). Additionally, a single isolate from each of the following rarer species was identified: *Candida lusitaniae*, *Candida dubliniensis*, *Trichosporon asahii*, and *Candida metapsilosis*. Notably, to the best of our knowledge, this is the first report of *C. metapsilosis* in Morocco.

Only by molecular techniques could we differentiate between highly related species pairs such as *C. albicans* and *C. dubliniensis*, since they both can form germ tubes and they both display a green color in CHROMagar. Moreover, the distinction between *C. parapsilosis* and *C. metapsilosis* proved elusive when relying on phenotype-based methods. Another noteworthy incongruence in identification arises with *T. asahii*. This particular species has been wrongly identified as *C. albicans* when utilizing CHROMagar, highlighting the limitations of relying only on phenotype-based methods.

With regard to the clinical origin of the sample, vulvovaginal candidiasis was the top fungal infection in our study, with 47 out of 93 cases (Table 1). Among these samples, *C. albicans* was the most represented, with 38 isolates (80.9%), followed by *C. glabrata* (6, 12.9%) and *C. parapsilosis*, *C. tropicalis*, and *C. kefyr* with a single isolate each (2.1%).

Stool was the second most common clinical source, with 20 isolates (21.5%), and included a wide variety of isolated species. Urinary infections (candiduria) were the third most common clinical source in this study, also characterized by the predominance of *C. albicans* with 10 isolates (62.5%). Overall, *C. albicans* was present in seven out of the nine isolation sources considered (all except nail and skin), followed by *C. glabrata* (found in vagina, stool, urine, and skin), *C. parapsilosis* and *C. tropicalis*, both in three sites (vagina, stool, and nail), and *C. kefyr* (vagina, stool), while all the other species were found in a single specimen type (Table 1).

### 3.2. Antifungal Susceptibility Testing

We next investigated the antifungal susceptibility of the collected strains using the EUCAST protocol. We focused on susceptibility to fluconazole and anidulafungin, two broadly used antifungals belonging to the two major antifungal classes: azoles and echinocandins, respectively. The MIC_50_ values of the strains exhibited variation, with the majority being susceptible at low concentrations according to EUCAST breakpoints for fluconazole (Table 2). However, it is important to note that two *C. albicans* isolates and one *C. glabrata* isolate exhibited intermediate resistance to this antifungal, with MIC_50_ values of 4 µg/mL and 16 µg/mL, respectively. The two strains of *C. krusei* showed high values of MIC_50_ towards fluconazole (8 and 16 µg/mL), indicative of their intrinsic resistance to this antifungal.

As for anidulafungin (Table 3), one *C. albicans* and one *C. tropicalis* isolate exhibited resistance to this drug according to EUCAST resistance breakpoints, as their MIC_50_ was higher than 0.03 and 0.06 µg/mL, respectively. In addition, 11 strains of *C. albicans* (2), *C. glabrata* (5)*,* and *C. parapsilosis* (4) were close to their respective resistance thresholds for this antifungal. This result is noteworthy as it indicates that around 11.8% of all strains surveyed have a reduced susceptibility to anidulafungin. Interestingly, the two isolates of *C. krusei* showed elevated values of MIC_50_ for this antifungal (0.25 and 4 µg/mL). With this finding, the concept of multidrug resistance emerges, as this species is already known by its intrinsic resistance to fluconazole. Furthermore, *T. asahii* is also known for its innate resistance to anidulafungin, which explains the increased MIC_50_ value obtained for this fungus (=8 µg/mL).

## 4. Discussion

Morocco bears a high burden of globally significant fungal diseases but has lagged behind in epidemiological data collection and reporting. Only few epidemiological data on fungal infections, including candidiasis, are documented in the country [7]. Recently, the World Health Organization (WHO) released a list of fungal priority pathogens in which two *Candida* species (*C. albicans* and *C. auris*) are considered in the critical priority group, and three others, *C. glabrata*, *C. parapsilosis,* and *C. tropicalis*, are in the high priority group [1]. However, knowledge on their relative incidence or resistance profiles in Morocco is very scarce. Hence, additional efforts are needed to close the large data gaps of both disease incidence and antifungal resistance at regional, national, and international levels. Here, we present the results of the first study conducted in the northwest region of Morocco to assess the relative prevalence of *Candida* species and to determine their antifungal susceptibility profile.

The methodology for the diagnosis of *Candida* infections in Morocco varies from phenotypic (germ tube, culture on chromogenic agar) [11], biochemical (API, yeast ID cards) [12], and immunological (Krusei color test, Bichrolatex albicans) [13] to molecular methods (PCR and Sanger sequencing) [14]. Phenotype-based tools are the main assays used for diagnostics in the country, as they are cost- and time-saving, as well as easy to perform. The clinical identification carried out by the participating laboratories of this study was based on the germ tube test. However, this test has important limitations that make it less reliable in comparison to other identification tools. The misinterpretation of elongated blastoconidia can give false positive or negative results, usually associated with inappropriate incubation conditions [15]. False positive results can be explained by confusing early formed pseudohyphae produced by some species such as *C. tropicalis* and *C. kefyr* due to increased incubation time [16,17,18]. On the other hand, heavy fungal inoculum (>10^7^ CFU/mL) can inhibit the production of germ tubes, leading to false negative results [15].

CHROMagar Candida provides a selective identification method for clinically relevant yeasts. Specific enzymes produced by *Candida* species react with chromogenic β-glucosaminidase substrate, resulting in the formation of colonies with distinct colors. The shift from using germ tube to CHROMagar identification revealed better specificity compared to the germ tube test (Figure 1). However, pairs of highly related species such as *C. albicans/C. dubliniensis* and *C. parapsilosis/C. metapsilosis* could not be differentiated using this media as their colonies exhibited the same color. CHROMagar identification for these cases is still controversial [19,20]. We observed that *T. asahii* and *C. albicans* rendered colonies with the same color, and a previous study [21] demonstrated that *C. krusei* could also be confused with *Candida firmetaria* and *Candida inconspicua* using this medium.

It was only by using Sanger sequencing of the ITS region that we could obtain more accurate and specific results. This not only revealed identification errors resulting from morphological approaches but also unearthed the presence of rare species, which would have been otherwise overlooked. This includes a case of *C. metapsilosis*, which becomes the first identification of this rare hybrid pathogen in Morocco [22]. This isolate came from a stool sample of a 3-year-old boy who suffered from a bacterial infection (symptoms were mainly loose stools with presence of mucus and high level of leukocytes). *C. metapsilosis* was likely not the reason for infection, since there were only few colonies in the isolation medium.

Vulvovaginal candidiasis was the most prevalent infection related to *Candida* species, comprising more than 50% of the cases in our study. This is consistent with previous studies. For instance, a study in Rabat (Morocco) considering 114 vaginal swabs found that *C. albicans* was the predominant species, present in 69.2% of the swabs, followed by *C. glabrata* and *C. tropicalis* with equal frequencies (15.5%) [23]. Similar results were also found with 697 samples [12]. In this study, the identification of *C. albicans* strains was based on morphological methods, while the non-*albicans* were classified using both biochemical (Api 20C AUX) and molecular tools (ITS sequencing). The identification results were summarized as follows: *C. albicans* (47.22%), *C. glabrata* (34.21%), *C. tropicalis* (3.94%), *C. parapsilosis* (3.94%), and *C. dubliniensis* (2.63%).

Stool and urine isolates were ranked as the second and the third clinical source, respectively, in our study, with stool samples having a notable species diversity and with *C. albicans* being the most frequently encountered species in both sources (40% and 62,5%, respectively). Recently, a retrospective study using BD Phoenix^TM^ Yeast ID (Becton Dickinson, NJ, USA) conducted in the eastern region of Morocco showed that yeast infections (109 urine samples) were mainly related to *C. albicans* (56.88%) [11]. Other *Candida* species were distributed as follows: *C. glabrata* (16.51%), *C. tropicalis* (11.01%), *C. parapsilosis* (6.42%), *C, lusitaniae* (2.75%), *C. krusei* (1.83%), and *C. kefyr* (0.92%). There is no published data related to *Candida* species isolated from stool samples in Morocco so far.

*Candida albicans* continues to be the most prevalent species associated with candidiasis worldwide, with recurrent vulvovaginal candidiasis as the most common associated infection. Similar results were obtained for invasive candidiasis in Spain [24], Portugal [25], and Belgium [26] using ITS sequencing.

Azole antifungals are considered the first-line treatment for Candidiasis in Morocco due to their broad-spectrum activity, good tolerability, and affordability compared to other antifungal drugs [12,27]. However, the increasing prevalence of azole resistance poses a threat. In our research, some isolates of *C. glabrata* showed intermediate susceptibility to the two tested antifungals based on EUCAST classification. Although they did not show full resistance, their susceptibility profile indicates a diminished sensitivity to the conventional antifungal drugs and raises concerns about a potential for future resistance development. *C. glabrata* exhibits inherently low susceptibility to azole derivatives, necessitating higher concentrations of fluconazole for effective inhibition compared to other *Candida* species [28]. In such cases, alternative antifungals like voriconazole and posaconazole [29,30] may be used, particularly for infections caused by intrinsically resistant species such as *C. krusei*, which is not susceptible to fluconazole.

Amphotericin B, while effective against *Candida* infections, is limited by its nephrotoxicity [31]. Despite the development of lipid formulations aiming to reduce its side effects, amphotericin B continues to be excessively costly and may pose affordability challenges for low-middle income countries [32]. In Morocco, it is predominantly used for treating nosocomial infections [33,34].

Echinocandins are the most recent class of antifungals [35,36]. These drugs target the synthesis of β-(1,3)-D-glucan, a key component of *Candida* fungal cell wall. Resistance to these antifungals primarily occurs through the emergence of point mutations within the conserved regions of FKS1 and FKS2 genes [37]. These mutations can lead to modifications in the target enzyme of echinocandins (β-1,3-glucan synthase), rendering the drug less effective in inhibiting fungal cell wall synthesis and, as a consequence, promoting the development of resistance in *Candida* species. An interesting observation from our study is that the two isolated strains of *C. krusei* were resistant to anidulafungin and fluconazole, representing a case of multidrug resistance. Global surveillance programs, such as ARTEMIS and SENTRY, have yielded an immense volume of valuable data concerning global trends in candidiasis [38,39]. GLASS (Global Antimicrobial Resistance Surveillance System) is a novel surveillance program that focuses on fungi as well as other microbial pathogens. It has initiated a worldwide collaborative effort to compile data on antifungal-resistant infections [40]. These surveillance programs can provide early detection of resistant strains, guiding antifungal treatment to enhance patient outcomes as well as encouraging the rational use of currently available antifungals to reduce the development of resistance, thereby preserving their efficacy. Furthermore, they offer valuable perspectives on the geographical variation in species occurrence, the distribution across different types and age groups, and alterations in the antifungal susceptibility observed in collected *Candida* isolates. In Morocco, the lack of epidemiological data has hindered participation in global antifungal surveillance programs, which provide invaluable insights into emerging fungal pathogens and aids in the provision of effective antifungal treatments. To improve diagnostic accuracy, a combination of phenotypic and molecular methods is necessary, along with the recommendation to conduct further molecular assays in clinical diagnosis.

The discovery of resistant and borderline-sensitive strains raises concerns about multi-resistance to antifungals. To address this, there is an urgent need for more accurate tools and preventive strategies to fight *Candida* fungal infections. Exploring alternative treatment options, like different antifungal drugs or combinations, is important.

Our findings emphasize the urgent need to tackle the rise of resistant *Candida* species, highlighting the importance of health authorities raising public awareness regarding the looming threat. Correlating fungal identification with susceptibility measurements is essential to establish well-organized profiling of *Candida* clinical strains, especially in the light of the increasing prevalence of non-albicans *Candida* species and their growing resistance to antifungal drugs.

## Figures and Tables

**Figure 1 jof-10-00373-f001:**
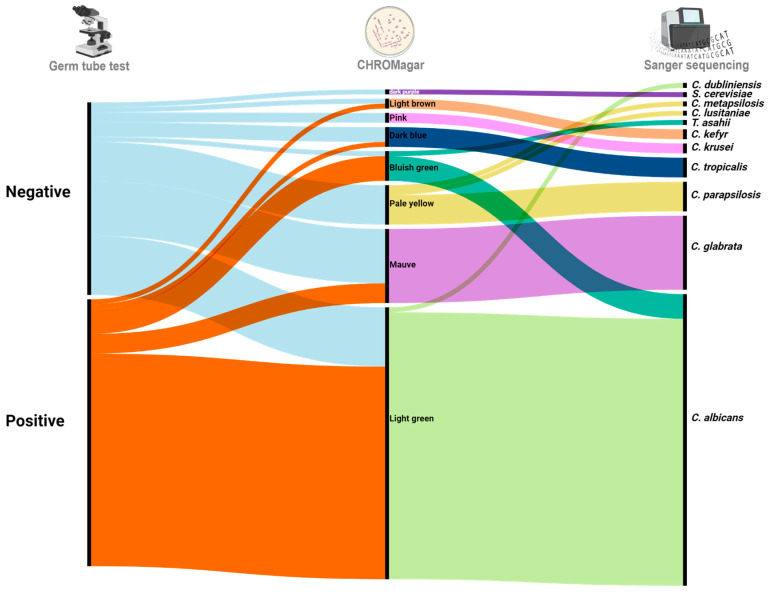
Phenotypic and molecular methods used for the identification of the fungal clinical species. The three vertical lines represent results from the three main approaches used; from left to right: Germ tube test can classify the collected fungal species into only two groups: producers of germ tubes (GTT+) and non-producers (GTT−). Using CHROMagar, the identification gets more specific as we can differentiate between species based on their colony color. Colors in the right side of the panel correspond to the colony morphology colors in the CHROMagar. Finally, PCR/Sanger sequencing provides more specificity. Lack of total congruence between morphological and molecular methods is illustrated by some of the strains presenting the same color in CHROMagar but they are actually different as observed using molecular tools.

**Table 1 jof-10-00373-t001:** Distribution of the fungal clinical species (n = 93) based on their source of isolation and molecular identification results.

	Clinical Source										
	Vagina	Stool	Urine	Nail	Urethra	Tongue	Sputum	Skin	Semen	Total	Percentage
*C. albicans*	38	8	10		1	1	1		1	60	64.52
*C. glabrata*	6	2	5					1		14	15.05
*C. parapsilosis*	1	2		3						6	6.45
*C. tropicalis*	1	2		1						4	4.30
*C. krusei*		2								2	2.15
*C. kefyr*	1	1								2	2.15
*C. metapsilosis*		1								1	1.08
*C. lusitaniae*			1							1	1.08
*C. dubliniensis*		1								1	1.08
*S. cerevisiae*		1								1	1.08
*T. asahii*				1						1	1.08
Total	47	20	16	5	1	1	1	1	1	93	100

**Table 2 jof-10-00373-t002:** Antifungal susceptibility of the clinical isolates to fluconazole.

	MIC 50 Values of FLC (µg/mL)								EUCAST Clinical Breakpoints (µg/mL) and Interpretation		
	0.25	0.5	1	2	4	8	16	Total	S≤	I	R>
*C. albicans*	51	4	1	2	2 *			60	2	4	4
*C. glabrata*				2	7	4	1 *	14	0.001	≤16	16
*C. parapsilosis*	5	1						6	2	4	4
*C. tropicalis*	1	1	1	1				4	2	4	4
*C. krusei*						1	1 **	2	ND	ND	ND
*C. kefyr*		1			1			2	ND	ND	ND
*C. metapsilosis*	1							1	ND	ND	ND
*C. lusitaniae*	1							1	ND	ND	ND
*S. cerevisiae*					1			1	ND	ND	ND
*C. dubliniensis*	1							1	2	4	4
*T. asahii*		1						1	ND	ND	ND
								93			

The tested concentrations of this antifungal ranged from 0.25 to 16 µg/mL. The samples of each species were distributed according to the MIC_50_ values derived from EUCAST clinical breakpoints: S = sensitive, I = intermediate, R = resistant. * Strains with intermediate resistance to fluconazole. ** *C. krusei* is intrinsically resistant to this antifungal. ND = Not determined.

**Table 3 jof-10-00373-t003:** Antifungal susceptibility of the clinical isolates to anidulafungin.

	MIC 50 Values of ANI (µg/mL)								EUCAST Clinical Breakpoints (µg/mL) and Interpretation	
	0.015	0.03	0.06	0.25	2	4	8	Total	S≤	R>
*C. albicans*	57	2 *	1 **					60	0.03	0.03
*C. glabrata*		9	5 *					14	0.06	0.06
*C. parapsilosis*					2	4 *		6	4	4
*C. tropicalis*	1	2		1 **				4	0.06	0.06
*C. krusei*				1 **		1 **		2	0.06	0.06
*C. kefyr*			1	1				2	ND	ND
*C. metapsilosis*					1			1	ND	ND
*C. lusitaniae*				1				1	ND	ND
*S. cerevisiae*			1					1	ND	ND
*C. dubliniensis*	1							1	ND	ND
*T. asahii*							1 ***	1	ND	ND
								93		

Anidulafungin was tested at concentrations ranging from 0.015 to 8 µg/mL. The samples of each species were distributed according to the MIC_50_ values derived from EUCAST clinical breakpoints: S = sensitive, R = resistant. * Strains close to the resistance breakpoint of anidulafungin, ** resistant strains to anidulafungin. *** *T. asahii* is intrinsically resistant to this antifungal. ND = Not determined.

## Data Availability

The original contributions presented in the study are included in the article, further inquiries can be directed to the corresponding author.

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
