# Peer review of "Assessing Diagnosis of Candida Infections: A Study on Species Prevalence and Antifungal Resistance in Northern Morocco"

_jof, 2024, doi:10.3390/jof10060373_

Round 1
Reviewer 1 Report
The manuscript Optimizing Diagnosis of Candida infections: A Study on Species Prevalence and Antifungal Resistance in Northern Morocco, by Ahaik et al. approaches an important matter - the reduced diagnosis capabilities.
On a first read through it is very interesting that there are no C krusei identified, although it is a very common strain. It is a nice work, however I have a few comments:
-lines 74-84: how where the strains selected to be included in the collection? - only 93 samples over an year from four labs is a small number
Not very clear how would this study optimize the diagnosis of fungi. It does not bring a new method, nor a cheaper or more accessible one. Since at the beginning of the article it is stated the problem, it does not seem to have a resolution by the end - an optimisation of diagnosis. I do not think that the authors are proposing to have sequencing for all identified strains on a daily basis. I recommend you change the title at least.
- change reference style to MDPI format
- check typos and punctuations; there are multiple errors
- subsection 2.3 - add a table with pictures of different colony morphologies (pigment) and their associated strain name
- line 146 - colonies of green colour can also be represented by C dubliniensis
Author Response
Responses to Reviewer 1
Major:
The manuscript Optimizing Diagnosis of Candida infections: A Study on Species Prevalence and Antifungal Resistance in Northern Morocco, by Ahaik et al. approaches an important matter - the reduced diagnosis capabilities.
Response: We thank the reviewer for his/her appreciation of our work
On a first read through it is very interesting that there are no C krusei identified, although it is a very common strain.
Response: We did identify two Candida krusei strains. As shown in the results section, they are unequivocally identified with the ChromAgar test and later confirmed with the ITS Sanger sequencing. 2/93 (~2%) is in line with reports in studies of global incidence ranging from 1 to 3% (see Clin Microbiol Infect 2014; 20 (Suppl. 6): 5–10)
It is a nice work, however I have a few comments:
-lines 74-84: how where the strains selected to be included in the collection? - only 93 samples over an year from four labs is a small number
Response: We agree this is not a big number, the four laboratories involved participated on a voluntary basis and were not large hospitals. Nevertheless, as there was no prior selection of strains, this set is representative.
Not very clear how would this study optimize the diagnosis of fungi. It does not bring a new method, nor a cheaper or more accessible one. Since at the beginning of the article it is stated the problem, it does not seem to have a resolution by the end - an optimisation of diagnosis. I do not think that the authors are proposing to have sequencing for all identified strains on a daily basis. I recommend you change the title at least.
Response: We agree with the reviewer in that the current title may confound the reader, we do not propose a solution but highlight what is missed with current approaches. We have changed optimising to assessing. Indeed, our research does not introduce a new method or significantly simplify accessibility or cost. Instead, it explores how molecular methods can complement existing diagnostic approaches to enhance accuracy, particularly when dealing with rare fungal species or those that are highly related. We consider that the change of title makes this clear and thank the referee for pointing this out.
Minor:
- change reference style to MDPI format
Response: We have adapted the reference style to MDPI
- check typos and punctuations; there are multiple errors
Response: We apologise for that, we have extensively revised the text and corrected several errors.
- subsection 2.3 - add a table with pictures of different colony morphologies (pigment) and their associated strain name
Response: The colours of different strains are indicated in Figure 1, now we specify this in the figure legend and the text.
- line 146 - colonies of green colour can also be represented by C dubliniensis
Response: Yes, but the point is that the manufacturer’s instructions do not mention that. (see https://www.chromagar.com/product/chromagar-candida/ )
Reviewer 2 Report
The study describes the Northern Morocco region, region which previously had only a little global information on fungal infections. Thus, the present study fills a gap in epidemiological data that is important for the treatment strategies within the region. Also, By investigating the prevalence of different Candida spp. and their patterns of resistance to commonly used antifungal drugs, the research helps to comprehend the changing dynamics of fungal pathogens. I appreciate the design of the study design, which involved collection of clinical isolates for a year and, therefore, it has created a rich information base that has made its results very dependable.
Although the study is good at identifying and grouping prevalent Candida spp. adding less common species would increase its comprehensiveness regarding the fungal domain in that area. What is more, future research could be improved by taking a longitudinal perspective that examines how resistance towards antifungal drugs and prevalence of species change across time. Moreover, the cross-regional comparison would assist in locating these results in a wider worldwide context.
Necessary
l. 152-167: clarify the molecular identification method confirming misidentification of some strains via phenotypic means.
l. 186-190: details on fluconazole and anidulafungin MIC50 values’ distribution to each of species should be given. Also, re-examine strain classification in terms of susceptibility, intermediates and resistant to determine if it agrees with the present EUCAST directives.
Fig. 1. caption: make the definition or legend of the figure more detailed to reveal the observed differences between morphological and molecular identification methods.
Desirable
l. 154-160: include a detailed listing of the 11 species from the 93 isolates in form of a table or additional text to know the distribution of species precisely.
l. 106-117: specify PCR conditions, please include buffer compositions and cycle thresholds, to ensure reproducibility by other studies.
Table 1: please add patterns of resistance, in addition to anti-microbial susceptibility testing, to the data; resistance findings on the most prevalent microorganisms will help to complete the picture.
Author Response
Responses to Reviewer 2
Major
The study describes the Northern Morocco region, region which previously had only a little global information on fungal infections. Thus, the present study fills a gap in epidemiological data that is important for the treatment strategies within the region. Also, By investigating the prevalence of different Candida spp. and their patterns of resistance to commonly used antifungal drugs, the research helps to comprehend the changing dynamics of fungal pathogens. I appreciate the design of the study design, which involved collection of clinical isolates for a year and, therefore, it has created a rich information base that has made its results very dependable.
Response: We thank the reviewer for his/her appreciation of our work
Although the study is good at identifying and grouping prevalent Candida spp. adding less common species would increase its comprehensiveness regarding the fungal domain in that area. What is more, future research could be improved by taking a longitudinal perspective that examines how resistance towards antifungal drugs and prevalence of species change across time. Moreover, the cross-regional comparison would assist in locating these results in a wider worldwide context.
Response: We managed to obtain a species level diagnosis of every strain in the collection, so if some rare species are missing it may be due to the relatively small sample size. We note, however, that we identified several rare Candida species such as C. metapsilosis, being our identification the first one for that species in the country. We totally agree with the reviewer in that future research should involve a longitudinal perspective and cross-regional comparisons, we hope our work sets the basis for such future research.
Necessary
- 152-167: clarify the molecular identification method confirming misidentification of some strains via phenotypic means.
Response: The molecular identification method is PCR amplification of the ITS region coupled to Sanger sequencing. This is explained in the materials and methods section. The misidentifications can be seen at Figure 1, which is now cited in that sentence.
- 186-190: details on fluconazole and anidulafungin MIC50 values’ distribution to each of species should be given. Also, re-examine strain classification in terms of susceptibility, intermediates and resistant to determine if it agrees with the present EUCAST directives.
Response: We have now included clinical breakpoints according to EUCAST as well as re-examined the strain classification according to the present EUCAST directives. All tables have been updated accordingly.
Fig. 1. caption: make the definition or legend of the figure more detailed to reveal the observed differences between morphological and molecular identification methods.
Response: We have now expanded the caption of figure 1.
Desirable
- 154-160: include a detailed listing of the 11 species from the 93 isolates in form of a table or additional text to know the distribution of species precisely.
Response: This is included in Table 1
- 106-117: specify PCR conditions, please include buffer compositions and cycle thresholds, to ensure reproducibility by other studies.
Response: We have revised the PCR conditions, they are detailed in the material and methods. The buffer was a commercial one, which is indicated. This is not a quantitative PCR so cycle thresholds were not used.
Table 1: please add patterns of resistance, in addition to anti-microbial susceptibility testing, to the data; resistance findings on the most prevalent microorganisms will help to complete the picture.
Response: Resistance and anti-microbial susceptibility testing and results are shown in tables 2 and 3, which have been modified to include breakpoints according to EUCAST.
Round 2
Reviewer 1 Report
The authors have performed all recommended changes.
No additional comments.